# Do deep neural networks possess concept space grid cells?

## Abstract

Place and grid-cells are known to aid navigation in animals and humans. Together with concept cells, they allow humans to form an internal representation of the external world, namely the concept space. We investigate the presence of such a space in deep neural networks by plotting the activation profile of its hidden layer neurons. Although place cell and concept-cell like properties are found, grid-cell like firing patterns are absent thereby indicating a lack of path integration or feature transformation functionality in trained networks. Overall, we present a plausible inadequacy in current deep learning practices that restrict deep networks from performing analogical reasoning and memory retrieval tasks.

## 1 Introduction

Cells in the hippocampal region of the rat brain signal the animal's location in space. This phenomenon has provided support to the idea that the rat hippocampus operates like a cognitive map [1]. Specific cells, termed as the "place cells" are known to selectively fire when the animal is in certain locations in the environment [1]. The discovery of place cells was followed by the discovery of "grid" cells, that selectively fire at multiple discrete and regularly spaced locations [2]. Extensive animal studies, especially in rats, involving electrophysiological recording have shown the existence of grid cells and place cells in localized brain structures, namely the medial entorhinal cortex and hippocampus respectively. In humans, the role of the medial temporal lobe and particularly the hippocampus in creation, consolidation and recall of declarative memories has been established [3]. Specific cells in the medial temporal lobe (MTL), called the "concept" cells, seem to fire when a particular concept is presented to the subject [4]. These findings established that humans are indeed capable of forming an internal representation of the external world. Navigating this internally represented abstract concept space might indeed form the basis of memory retrieval or context-driven reasoning.

Deep learning (DL) has brought about a new dawn in artificial intelligence by enabling models to learn representations of data with multiple levels of abstraction. However, most of these models lack the flexibility to perform relational reasoning. These issues have rekindled interest among researchers to understand and replicate the brain's learning process. The success of deep neural networks (DNN) is generally attributed to their ability to identify optimal discriminative features in a given dataset. This learnt feature space can be thought of as a "concept" space for the DNN. The ability to perform relational reasoning depends on the network's ability to navigate over this concept space. To solve analogical problems, the network needs to apply the desired feature transformation to arrive at the correct solution. For instance in the image space, the network needs to navigate the visual concept space to understand that an $OR$ operation between a "dog image" and a "ball image" could lead to an image of "a dog playing with a ball".

The ability to navigate the concept space depends on the properties of the constituent neurons. Akin to memory retrieval or context-driven reasoning in humans, DNN would need neurons that show

place and grid-cell like activation properties. Artificial neurons that respond to specific classes of images could be considered as showing place cell-like activity in the concept space. Consequently, all final layer neurons of a DNN trained for a classification problem would be showing place-cell like activity. However, the ability to navigate the concept space relies on the DNN's ability to perform "path integration", or the ability to localize its position in the concept space given specific feature transformations. Grid cells are known to be responsible for path integration in rodents and humans. Therefore, the ability of the DNN to solve analogical problems relies on the presence of grid cell-like properties among DNN neurons.

In this work[1], we aim to investigate a particular class of DNN, namely convolutional neural network (CNN). We investigate the firing properties of neurons, specifically in the final and pre-final layers, to understand their activation patterns while the network performs classification.

## 2   Methods and Results

We trained a CNN to identify hand-written digits $0 - 9$ in the MNIST dataset. The dataset has 60,000 images for training and the network was tested on 10,000 images. The network architecture consists of two 2D convolutional layers (filter size 5) followed by 2D Max-pooling layers. The output of the convolutional layers was fed to a dropout layer and further input to two fully connected layers with 50 and 10 hidden units. Rectified Linear Unit (ReLU) was used as the activation function. The network was implemented on PyTorch 1.1.0 and trained to optimize negative log-likelihood loss using the Adam optimizer. The network was trained using a learning rate of 0.001 for 10 epochs (training batch size = 128). At the end of 10 epochs, the network's accuracy on the testset was **98.79 %**.

Once the network was trained to yield a satisfactory classification accuracy on the testset, we investigated the activation properties of constituent neurons. We limited our analysis to the pre-final and final layers (Layer 6 & 7 respectively) only. To visualize the concept space, we used the t-distributed Stochastic Neighbourhood Embedding (tSNE) [5] to generate a low-dimensional representation (2D representation here) of the feature space. We explored two concept spaces – one generated from the tSNE plot of raw image data and the other generated from the activation of the final layer neurons of the network. The first corresponds to the image concept space and the second corresponds to the concept space learned by the network, also referred hereafter as the network concept space. We generated the aforementioned spaces for the testset images (10,000 points). Figure 1 shows the two concept spaces.

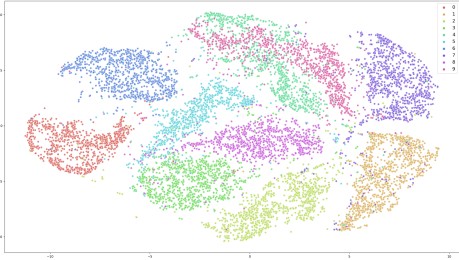 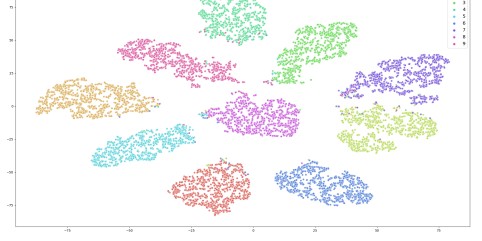

(a) tSNE representation of the image space for the MNIST testset.

(b) tSNE representation of the learned feature space for the MNIST testset.

Figure 1: Concept spaces obtained from tSNE representations of higher dimensional data in 2 dimensions.

To observe the firing property of the neuron across the concept space, the tSNE plots were color-coded using the respective neuron's activation values. Figure 2 shows the firing pattern of Layer 7 neurons. All the final layer neurons showed place-cell like activity, with preference to one cluster. Given the training paradigm, this was expected. Further, we extended our analysis to observe the firing pattern of the previous layer.

The pre-final layer had 50 neurons. However, none of the neurons had grid-cell like firing property. We used Principal Component Analysis (PCA) and obtained 11 PCs that significantly explained the

---

[1]Code and results available here

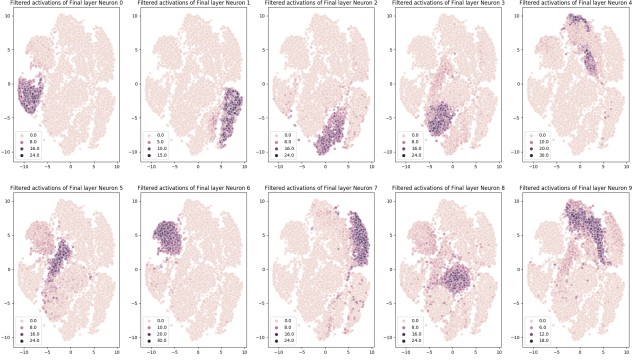

(a) Activation patterns of final layer neurons in image space

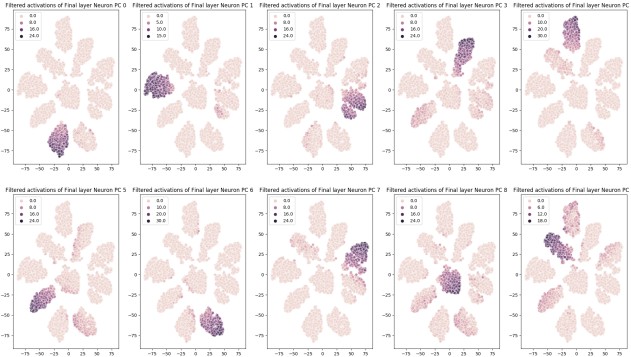

(b) Activation patterns of final layer neurons in network space

Figure 2: Activation patterns of the final layer neurons represented using color coding on the tSNE plots. All non-negative activities are clamped to 0 as they correspond to highlight regions with high neuronal activation, akin to firing rates.

firing properties of these neurons. The firing patterns of these 11 PC neurons are shown in Figure 3. Since the firing patterns of the PC neurons are obtained from a linear combination of those of the pre-final neurons, the absence of a grid-like firing pattern in the 50 neurons is reflected in the firing patterns shown in Figure 3. Instead these neurons show preferential firing to multiple clusters, thus indicating that these neurons respond to images of multiple digits.

## 3   Discussions & Conclusion

In this work, we explored the firing patterns of final and pre-final layer neurons in a CNN trained to identify hand-written digits in the MNIST dataset. Although the final layer neurons showed place-cell like firing patterns in the learned concept space, the pre-final layer neurons failed to show any grid-cell like firing patterns. The distinction in firing patterns is clearer in the network space as compared to the image space, thus indicating that the network coded for certain regions in the concept space. However, the absence of grid-cell like firing indicates that the network failed to learn the transformations and manifolds in the concept space. This would imply that the network would be unable to navigate this space and thereby perform feature transformation operations in this space. Thus, the trained CNN would fail to perform analogical problems, similar to other deep models trained on more complicated datasets.

The network analyzed here was a fairly simple and shallow network as compared to practical deep networks. However, it serves as a toy example to illustrate the inadequacy of current training practices that restrict deep networks from performing analogical reasoning tasks. Since the images were shown to the network in isolation, the network was unable to learn the information of manifolds and

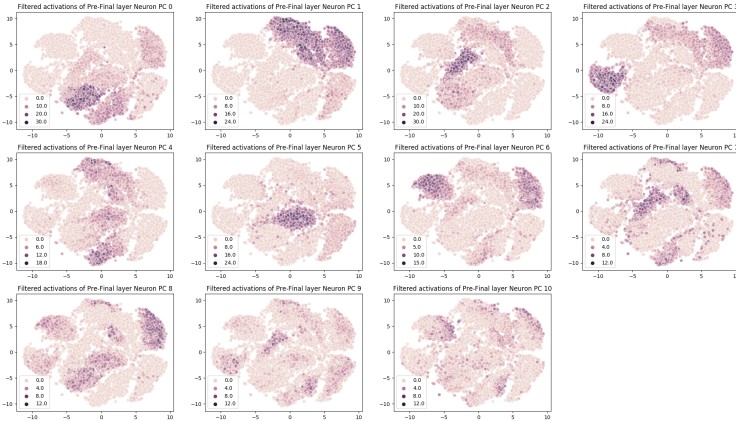

(a) Activation patterns of the principal components of pre-final layer neurons in image space

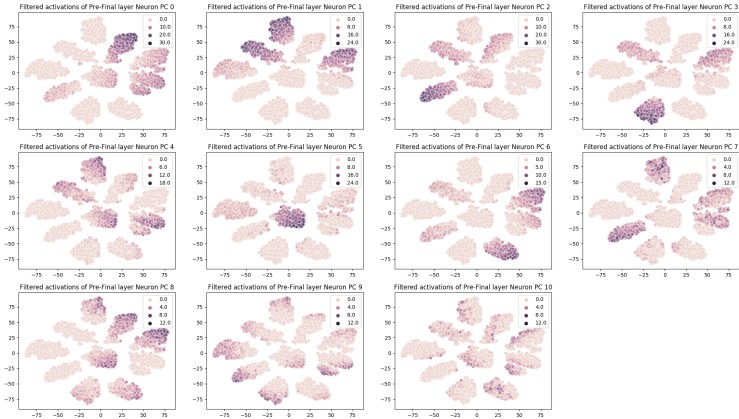

(b) Activation patterns of the principal components of pre-final layer neurons in network space

Figure 3: Activation patterns of the principal components of pre-final layer neurons represented using color coding on the tSNE plots.

relative transformation among the clusters. Adding examples that lie on manifolds in this space could alleviate this problem. More recently, Hill et al. proposed a framework to incorporate analogical reasoning in deep networks using a modified training strategy [6]. It would be interesting to observe the firing patterns of neurons in such a network, as done here, to inspect if the network navigated the concept space similar to the animal brain.

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
