# OpenReview forum: "Do deep neural networks possess concept space grid cells?"
_NeurIPS.cc/2019/Workshop/Neuro_AI — Submitted to Real Neurons & Hidden Units @ NeurIPS 2019_

### Official Review · AnonReviewer2 · 2019-09-24
**Neural networks trained on pure MNIST classification don’t appear to show grid representations**

**Clarity:** 3

**Category:**

Common question to both AI & Neuro

**Clarity Comment:**

Overall the writing is relatively clear, but it would have been beneficial to describe the hypotheses more explicitly, e.g. what neural activity would be expected for a place, grid, or concept representation with respect to MNIST.

**Evaluation:**

2: Poor

**Importance:**

2: Marginally important

**Importance Comment:**

While the question of how neural networks may act over concept space is important, I don’t think the approach used by the authors correctly adress this question. The work of Hill et al. (2019) very clearly addresses these questions by devising tasks that require generalization across domains, showing how training regime is sufficient to overcome the difficulties of these tasks, even in shallow networks. I don’t see how the current work adds more clarity to this research direction.

**Intersection:**

4: High

**Intersection Comment:**

The question of how the brain and artificial network can perform relational reasoning is critical in both fields, since many believe that it may be one of the primary ingredients of intelligence. It’s also critical to understanding the function of the hippocampus and entorhinal cortex in humans.

**Rigor Comment:**

The main point relies purely on a visual representation of the top PCs of the penultimate layer of a CNN, which I believe is insufficient. The authors should have identified a task where networks trained on MNIST perform poorly, and then propose a different strategy or architecture.

**Technical Rigor:**

1: Not convincing

---

### Official Review · AnonReviewer3 · 2019-09-25
**Potentially interesting idea, but over-interpreted results**

**Clarity:** 3

**Comment:**

- I am not convinced that the ability of deep networks to solve analogical problems relies on the presence of grid-like properties in the hidden units. Perhaps this is ignorant of me, but I think that this is a critical point for the paper and thus needs to be better motivated and explained. In particular, I think that while grid cells can support path-integration, not all networks that path-integrate necessarily contain cells with hexagonal symmetry.
- I am not surprised that the network trained by the authors does not show grid-like responses. It seems reasonable that the network learned to classify each number separately, without learning the full manifold. If someone were to record from a real brain from the visual areas while the animal was performing a discrete visual classification task, I am not sure that they would see grid cells there either. Thus, unfortunately the current paper reads that the authors trained a network that didn't need to learn the full manifold, so it didn't, and then didn't show properties that one may (or may not) expect for it to exhibit if it had learned the full manifold. I think there could be something interesting in this endeavor, but the implementation carried out by this paper wasn't very convincing to me.
- A minor (but important) comment - grid cells have not yet "known to support path integration in rodents and humans", since there is no casual experiment that shows their necessity. I think this statement is also indicative of my general complaint - that the importance of grid cells is not fully fleshed out or supported in this work, and thus the lack of grid cells in networks is over-interpreted.


**Category:**

AI->Neuro

**Clarity Comment:**

The paper was overall relatively well-written and easy to follow. The figures were simple and easy to interpret. The authors made their methods, results, and claims quite clear.

**Evaluation:**

2: Poor

**Importance:**

2: Marginally important

**Importance Comment:**

I do think that investigating under what conditions in artificial networks grid cells appear is very interesting. However, I was not fully convinced that the results presented here made substantial contributions to the AI or neuroscience field.

**Intersection:**

4: High

**Intersection Comment:**

The authors investigated neuroscience-informed properties of DNN. I think that searching for neuroscience-inspired properties in deep networks can be interesting, and is certainly within the intersection of AI and neuroscience.

**Rigor Comment:**

While the specific techniques employed by the authors seem perfectly fine and relatively rigorous, the question itself (do hidden units in the later layers contain grid-like patterns) felt rather simple and uninformative (simple is great, as long as it still tells you something interesting).

**Technical Rigor:**

2: Marginally convincing

---

### Official Review · AnonReviewer1 · 2019-09-26
**Interesting question, but significant problems with underlying assumptions and analysis methods**

**Clarity:** 2

**Comment:**

The authors propose training a deep network and seeing if activations similar to concept and grid cells exist in the hidden layers. The motivation, however, is strenuous, and the results are not convincing. It is unclear why these cells are necessary and sufficient to do relational reasoning as the authors claim, and the PCA / tSNE don't seem to address the question the authors sought to address.


**Category:**

Neuro->AI

**Clarity Comment:**

Design choice and analysis are poorly justified, with some ambiguity about how the results were obtained. It’s also unclear why the architecture, optimizers, dataset were chosen. Certain methods were also not justified - why limit the analysis to last two layers? Also why is figure 1 focused on the output layer, which are trained to represent the classes?
Not clear on why the plot ‘shows place-cell like activity’. what is the justification there? tSNE embedding space cannot be interpreted as it’s not linear. Similar problems apply to the PCA analysis. Grid cells represent cells that fire in response to various locations, not separate cells that fire to resemble some geometric pattern within the brain.

**Evaluation:**

1: Very poor

**Importance:**

2: Marginally important

**Importance Comment:**

While the general method of training neural networks and examining their representations for key insights and relations to neuroscience is a valid one, the methods and results do not seem to answer it in a way that is principled or well thought out. There is also substantial misunderstanding about neuroscience concepts throughout the paper.


**Intersection:**

3: Medium

**Intersection Comment:**

Aside from the obvious take, the intersection is somewhat strained. Unclear what the ultimate goal of the paper is - as the idea appears to be based on a flawed understanding of mechanistically what’s required in the brain to solve certain problems.


**Rigor Comment:**

There are major logic / misrepresentation issues throughout the paper, some based on incorrect assumptions and possible misunderstandings. The neuroscience motivation lacks consensus in the community and some concepts are wrongly characterized, such as the definition of path integration. This greatly weakens the motivation and connections to deep learning.

The main method is a simple analysis of vanilla CNN trained on MNIST classification. The results are fairly unconvincing and not substantially justified in the approach. For example, the justification that this won’t allow the network to navigate in concept space seems arbitrary. Finally, it implies that classification is the right task to train the network to perform tasks based on relational reasoning. What about unsupervised tasks, and training with other models such as VAEs?

**Technical Rigor:**

1: Not convincing

---

### Decision · Program_Chairs · 2019-10-01

**Decision:**

Reject

**Comment:**

Unfortunately, we had more submissions than we could accept and based on the review process, we have decided not to accept your submission.  Nevertheless, thank you for your submission and interest in our workshop.